# Identification of Functional Brassinosteroid Receptor Genes in Oaks and Functional Analysis of *QmBRI1*

**DOI:** 10.3390/ijms242216405

**Published:** 2023-11-16

**Authors:** Wanfeng Ai, Hanzhang Liu, Yutao Wang, Yu Wang, Jun Wei, Xiaolin Zhang, Xiujun Lu

**Affiliations:** 1College of Forestry, Shenyang Agricultural University, Shenyang 110866, China; wfai23@syau.edu.cn (W.A.);; 2Key Laboratory for Silviculture of Liaoning Province, Shenyang 110866, China

**Keywords:** *Quercus* L., functional BR receptors, BRI1 gene family, *BRI1-BRLs*, evolution

## Abstract

Brassinosteroids (BRs) play important regulatory roles in plant growth and development, with functional BR receptors being crucial for BR recognition or signaling. Although functional BR receptors have been extensively studied in herbaceous plants, they remain largely under-studied in forest tree species. In this study, nine BR receptors were identified in three representative oak species, of which BRI1s and BRL1s were functional BR receptors. Dispersed duplications were a driving force for oak BR receptor expansion, among which the Brassinosteroid-Insensitive-1 (BRI1)-type genes diverged evolutionarily from most rosids. In oak BRI1s, we identified that methionine in the conserved Asn-Gly-Ser-Met (NGSM) motif was replaced by isoleucine and that the amino acid mutation occurred after the divergence of *Quercus* and *Fagus*. Compared with *QmBRL1*, *QmBRI1* was relatively highly expressed during BR-induced xylem differentiation and in young leaves, shoots, and the phloem and xylem of young stems of *Quercus mongolica*. Based on *Arabidopsis* complementation experiments, we proved the important role of *QmBRI1* in oak growth and development, especially in vascular patterning and xylem differentiation. These findings serve as an important supplement to the findings of the structural, functional and evolutionary studies on functional BR receptors in woody plants and provide the first example of natural mutation occurring in the conserved BR-binding region (NGSM motif) of angiosperm BRI1s.

## 1. Introduction

Brassinosteroids (BRs) are a group of plant-specific steroidal hormones [1] that regulate various essential growth and developmental processes, including cell elongation and division, xylem differentiation, and photomorphogenesis [2]. In plant vascular development, cell division and xylem differentiation are crucial processes in vascular establishment (which combines growth and patterning) and differentiation, respectively [3]. These processes directly affect wood production in woody plants by producing more radial cells along the tree ring [4]. The effects of BR perception and signaling on vascular development have been described in herbaceous plants such as *Arabidopsis* and *Solanum lycopersicum* (tomato) [2,5]. Following BR perception at the plasma membrane by Brassinosteroid-Insensitive-1 (BRI1) and co-receptor kinase BRI1-associated kinase, a canonical cascade relays BR signals to BRI1-EMS-SUPPRESSOR 1/BRASSINAZOLE-RESISTANT 1 family transcription factors, which regulate plant vascular development by controlling BR-regulated gene expression [2,5]. Thus, as the main receptor of BRs, BRI1 plays an essential role in BR recognition or signaling [6,7].

In *Arabidopsis* and rice, BRI1 has three other homologues: BRL1, BRL2, and BRL3 (BRLs) [7,8]; however, only BRL1 and BRL3 are considered functional BR receptors [9]. Unlike the growth-promoting BRI1 pathway in most cells, BRL1/3 receptor signaling only functions under certain spatiotemporal constraints [9] such as vascular differentiation [7]. BRI1 and BRLs share similar protein structures: an N-terminal signal peptide, a leucine-rich repeats (LRR) supercoil with a terminal island domain (ID), a single transmembrane domain, and a C-terminal serine/threonine (Ser/Thr) kinase domain [7]. Of these, the ID together with the carboxy-terminal flanking LRR (ID-LRR) is the minimal BR-binding domain [6]. The Asn-Gly-Ser-Met (NGSM) motif is a highly conserved sequence only located in the ID-LRR domain of reported functional BR receptors and not in that of BRL2s [7,8,10,11,12,13,14]. It is an important marker in determining whether BR receptors have BR-binding abilities [10]. Therefore, BRI1 and BRLs must be accurately identified to better understand the functions of functional BR receptors within them. Currently, the conservation of the BR-binding region (NGSM motifs) of functional BR receptors has been validated in a variety of plants [7,8,10,11,12,13,14]. Meanwhile, genome information on herbaceous plants such as soybean and millet has enhanced the understanding of the structure, function, and evolution of functional BR receptor genes [11,15]. In woody plants, the genome-wide identification of poplar functional BR receptor genes and subsequent mutant studies have revealed the similarities and differences in the functions of poplar and *Arabidopsis* functional BR receptors [13]. However, it is largely unknown whether the classical BR-binding region in functional BR receptors are also conserved across slow-growing forest tree species, and what functions these receptors play in these species.

Oaks (*Quercus* L., Fagaceae) are among the most important forest tree species in terms of species diversity, ecological dominance, and economic value [16]. There are more than 450 oak species worldwide, primarily distributed in the temperate forests of the northern hemisphere, which constitute a critical global renewable resource and play important roles in carbon sequestration and water and soil protection [17]. Oak is a ring-porous species; the canonical differences in size between the earlywood and latewood vessels give its wood a beautiful texture [18]. Its wood is considered a high-quality raw material for timber products, possessing high economic value [19]. The genomes of seven oak species (including *Quercus robur* [16,20], *Q. lobata* [17,21], *Q. suber* [22], *Q. mongolica* [19], *Q. gilva* [23], *Q. variabilis* [24], and *Q. acutissima* [25]) have been completely sequenced, which has laid the foundation for our research into the characteristics of the classical BR-binding region of oak functional BR receptors, as well as the function and evolution of these receptors.

Here, we performed genome-wide identification of functional BR receptor genes in three representative oak species—*Q. mongolica*, *Q. lobata*, and *Q. suber*—and characterized their gene and protein structures, the BR-binding conserved region, and phylogenetic relationships. We also analyzed the expression patterns of functional BR receptors during BR-induced stem vascular differentiation and in different *Q. mongolica* vascular tissues. Furthermore, we conducted *Arabidopsis* complementation experiments to validate the important role of *QmBRI1* in growth and development, especially in vascular development.

## 2. Results

### 2.1. Identification of Functional BR Receptor Genes in Oaks

Seventeen putative BR receptor genes [*BRI1-BRLs*; *Q. mongolica* (3), *Q. lobata* (3), *Q. suber* (3), *Q. gilva* (3), *Q. acutissima* (3), and *Q. robur* (2)] were identified via the BLAST search of the six oak genomic datasets. In addition, using the same approach failed to identify BR receptor genes in the currently published *Q. variabilis* genomic data [24]. Based on the phylogenetic relationship of 21 *Arabidopsis* and oak *BRI1-BRLs* (Appendix A), these putative oak *BRI1-BRLs* were named following the nomenclature of *Arabidopsis*. They encoded proteins ranging from 858 to 1221 amino acids. Based on the available genome annotation information, the identified *BRI-BRLs* were found to be distributed across different chromosomes in their respective genomes (Appendix A). Specifically, *Q. mongolica*, *Q. lobata*, and *Q. robur BRI1-BRLs* were located on chromosome 6 (*QmBRI1* and *QlBRI1*), 8 (*QmBRL1*, *QlBRL1*, and *QrBRL1*), and 3 (*QmBRL2*, *QlBRL2*, and *QrBRL2*). In the case of *Q. gilva*, its *BRI1-BRLs* were observed on chromosome 8 (*QgBRI1*), 5 (*QgBRL1*), and 12 (*QgBRL2*). Lastly, *Q. acutissima BRI1-BRLs* were found on chromosome 8 (*QaBRI1*), 7 (*QaBRL1*), and 5 (*QaBRL2*). These findings enabled us to identify three paralogous genes in different oaks (except for *Q. robur* and *Q.variabilis*). Of these, the identified protein sequences of *BRI1-BRLs* in three oak species (*Q. mongolica*, *Q. lobata*, and *Q. suber*) were complete. The nine putative oak *BRI1-BRLs* were used for further bioinformatics analysis (Table 1). Collinearity analysis revealed that *Q. mongolica* and *Q. lobata BRI1-BRLs* were generated by the dispersed duplication. Gene structure analysis showed that none of the nine putative oak *BRI1-BRLs* coding regions contained introns (Appendix A). The protein sequences of the nine genes had a similar number of amino acids ranging from 1133 to 1221 and molecular weights ranging from 124.46 kDa to 132.42 kDa (Table 1). The similarity between BRI1-BRLs orthologs in the three oak species was high, exceeding 98%; they were all negatively charged stable proteins. Moreover, they all had a certain degree of hydrophilicity except for QlBRL1. All nine putative BRI1-BRLs were predicted to be in the cell membrane and contained an LRR domain with an ID, two conservatively spaced cysteine pairs, a signal peptide, a transmembrane domain, and a Ser/Thr kinase domain (Appendix A). We also identified 10 conserved motifs, of which motifs 1, 2, and 5 could be mapped to the LRR domain. Motifs 3–4 and 6–10 comprised the ID and Ser/Thr kinase domain, respectively (Appendix A). After comparison, we found that *BRI1-BRLs* in oaks and *Arabidopsis* were similar in gene and protein structure (Appendix A), verifying the homology between them. Compared with the reported and verified protein sequences of plant *BRI1-BRL1/3* genes, the methionine in the NGSM motif was replaced by isoleucine in the three oak BRI1s (Figure 1A), indicating that the canonical NGSM motif was not conserved in plant BRI1 proteins. It is noteworthy that this specific amino acid mutation observed in oak BRI1s was not detected in its ancestral species within the Fagaceae family, namely *Fagus sylvatica* (Appendix A). Additionally, none of the BRL2 proteins in the three oak species contained a canonical BR-binding region NGSM motif in their ID-LRR domains. This finding was consistent with that seen for *Arabidopsis* (Appendix A), suggesting that only the oak *BRI1* and *BRL1* genes, but not BRL2, are putative functional BR receptors and can bind to BR or restore BR responsiveness.

### 2.2. Evolution Analysis of Oak Functional BR Receptor Genes

To understand the evolution of oak functional BR receptors, we selected representative species to reconstruct the phylogenetic relationship of angiosperms *BRI1-BRLs* by considering the angiosperm phylogeny group classification (APG IV) [26] and available high-quality genomic data (Figure 1B). The Bayesian and ML (Appendix A) phylogenetic analyses of representative angiosperms revealed the presence of 48 *BRI1-BRLs*, which were divided into three canonical clades (Figure 1C). Clades I, II, and III represented BRI1-type, BRL1/3-type, and BRL2-type genes, respectively. The phylogenetic trees reconstructed by the two methods only showed a small difference in the bootstrap values of individual branches. With strong bootstrap value support in each clade, monocot and eudicot proteins were clustered into two separate groups. In the BRI1-type and BRL1/3-type clades, asterid and rosid proteins were further clustered into two subgroups, consistent with the taxonomic relationship among eudicots; oak proteins further diverged from *Arabidopsis* and poplar proteins only in the BRI1-type clade. These results collectively revealed that during angiosperm *BRI1-BRLs* evolution, BRI1-type genes were the most diverse. Moreover, oak BRI1s were closer to soybean BRI1s than to *Arabidopsis* and poplar BRI1s, suggesting possible functional differences between them.

### 2.3. Transcript Levels of Functional BR Receptors in Q. mongolica

To further understand the potential function of oak functional BR receptors, we used qRT-PCR to determine the *QmBRI1* and *QmBRL1* expression patterns in various *Q. mongolica* tissues and xylem differentiation. The *QmBRI1* and *QmBRL1* expression patterns overlapped, but not entirely in different tissues (Figure 2A). The relative expression levels of *QmBRI1* in young leaves, shoots, the phloem of the upper stems, and xylem of the upper stems were 13.1-, 8.1-, 4.6-, and 3.1-fold higher than those of *QmBRL1* (*p* < 0.05), indicating that *QmBRI1* may play a key role in young leaf, shoot, and stem development. We also measured the anatomical features of the basal stems 28 days after exogenous BL treatment (Figure 2B,C). The results showed that exogenous 1 μM BL application significantly increased the width and cell number of newly developed secondary xylem in *Q. mongolica* seedlings (Figure 2D,E) (*p* < 0.05), indicating that BR promoted xylem differentiation in *Q. mongolica*. We also examined *QmBRI1* and *QmBRL1* expression in the basal stems at different time points within 48 h of the same BL treatment (Figure 2F). Without BL treatment, the relative expression levels of *QmBRI1* were approximately 4-fold higher than those of *QmBRL*1 at different time points (*p* < 0.05), indicating that *QmBRI1* played a key role in stem development. Under 1 μM BL treatment, the relative expression levels of *QmBRI1* and *QmBRL1* were inclined to increase first and then decrease over time; this cycle repeated twice within 48 h; the relative expression of *QmBRI1* was significantly higher than that of *QmBRL1* at different time points (*p* < 0.05). Compared with no BL treatment, *QmBRI1* expression was significantly increased at all five time points after 1 μM BL treatment (*p* < 0.05), whereas *QmBRL1* expression was significantly increased at only three of these time points (*p* < 0.05). These results indicated that both *QmBRI1* and *QmBRL1* could respond to exogenous BR, with *QmBRI1* playing a more dominant and stable role in BR-induced xylem differentiation.

### 2.4. Subcellular Localization of QmBRI1

Previous reports have shown that AtBRI1 is located at the plasma membrane and endosome. To understand whether QmBRI1 has a similar localization pattern, we generated AtBRI1-GFP and QmBRI1-GFP fusion protein expression constructs. The produced constructs and the control pRI101-GFP vector were transiently expressed in *Arabidopsis* mesophyll cell protoplasts. The fluorescence was examined using confocal laser scanning microscopy. As expected, a QmBRI1-GFP signal was observed on the plasma membrane of *Arabidopsis* protoplasts, which was exactly the same as that of AtBRI1-GFP, indicating that QmBRI1 was also targeted to the plasma membrane in *Arabidopsis* (Figure 3A). This is consistent with our prediction.

### 2.5. Ectopic Expression of QmBRI1 Restores Growth Retardation in Bri1-5 Mutants

The *Arabidopsis bri1-5* mutant is a commonly used weak BR mutants [27]. It is in the Ws-2 ecotype background and shows noticeable growth retardation—plant dwarfism, shortened siliques and petioles, and shrunken and rounded leaves [28]. As a stable genetic transformation and regeneration system has not been established in *Q. mongolica*, we transformed the recombinant plasmid pRI101-QmBRI1-GFP into *bri1-5* to verify the biological function of *QmBRI1.* After preliminary growth analyses of three transgenic lines, two lines with relatively large phenotypic differences in leaves (*QmBRI1OX-1* and *QmBRI1OX-3*) were selected for subsequent growth complementation analysis (Appendix A). We investigated leaf characteristics and petiole growth 21 days after germination in the two *QmBRI1* overexpression lines, *bri1-5* mutants, and Ws-2 ecotypes. The overexpression lines and Ws-2 ecotypes had wider leaves (Figure 3B) and significantly longer petioles (*p* < 0.05) than the *bri1-5* mutants (Figure 3C). At 35 days after germination, we also measured the inflorescence stem and silique lengths at the same developmental position (Figure 3D,E). The *QmBRI1* overexpression lines and the Ws-2 ecotypes had significantly longer inflorescence stems and siliques than those of the *bri1-5* mutants (*p* < 0.05) (Figure 3F,G). Therefore, the stunted development of the mutant plants was restored when *QmBRI1* was expressed constitutively in *bri1-5*, revealing that *QmBRI1* acted as a functional ortholog of *AtBRI1* at multiple stages of seedling development.

### 2.6. Overexpression of QmBRI1 in the Bri1-5 Mutant Increases Vascular Bundle Numbers and Xylem Differentiation

The wild-type *Arabidopsis* inflorescence stem is a common model for studying wood development [29] because of its similar vascular pattern to that of most dicots [30]. The completion of primary provascular development can be observed at the base of its main inflorescence stem [31]. Procambial cells produce functional xylem and phloem, forming a vascular bundle (VB) and differentiated interfascicular fibers (Figure 4A) in between bundles [31]. The vascular cambium, which produces secondary vascular tissues, was evidently present at the basal ends of 5-week-old wild-type stems. The fascicular (Figure 4B) and interfascicular (Figure 4C) cambia are connected outside the sclerotic arc, forming a continuous ring of meristematic cells that primarily produce sclerified xylary tissue [30]. To investigate the potential functions of *QmBRI1* in vascular development, at 5 weeks after germination, we observed the anatomical structure of the inflorescences stem bases of the *bri1-5* mutant, Ws-2 ecotype, and one of the *QmBRI1*-overexpressing lines (Figure 4A–C). The *QmBRI1OX-3* line was selected as a representative of the *QmBRI1* overexpression lines because of its relatively strong growth phenotype. Sections collected from the bases of the stems of the *QmBRI1* overexpression lines and Ws-2 ecotypes shared anatomical similarities. However, all these plants had more VBs than the *bri1-5* mutants (Figure 4D), indicating the important role of *QmBRI1* in vascular patterning. Additionally, they produced more lignified tissue than the *bri1-5* mutants (Figure 4B,C). To further quantify the production of lignified tissue in the *bri1-5* mutants, Ws-2 ecotypes, and *QmBRI1OX-3* lines, we measured the radii of the fascicular xylem (indicated as a in Figure 4E), middle part of the interfascicular arc (b in Figure 4E), and bundle-flanking region (c in Figure 4E). At 5 weeks after germination, xylem formation in the bundles and interfascicular regions was significantly higher (*p* < 0.001) in the Ws-2 ecotypes and the transgenic plants than in the *bri1-5* mutants (Figure 4F–H); such a result was primarily attributed to the level of xylem differentiation in the fascicular and interfascicular cambia, further indicating that *QmBRI1* plays a key role in xylem differentiation. 

## 3. Discussion

In this study, three BR receptor genes were identified in oak species, of which *BRI1* and *BRL1* were functional BR receptors. The *Quercus* genus can be classified into two subgenera, namely *Quercus* (including species such as *Q. mongolica*, *Q. lobata*, and *Q. robur*) and *Cerris* (including species such as *Q. suber*, *Q. gilva*, *Q. variabilis*, and *Q. acutissima*) [19]. The variations observed in the chromosomal positions of *BRI1-BRLs* across different *Quercus* species may indicate the conservation of genomic evolution within the subgenus *Quercus* and the divergence in genomic evolution between the two main subgenera. The complete protein sequences of three *BRI1-BRLs* genes were only identified in three of the seven published oak genomes; such a limitation was caused by differences in genome assembly and annotation quality between different oaks [19]. The three oak species sampled in this study belong to the two main subgenera of *Quercus* and are found in three different continents: Asia, North America, and Europe [32]. Nonetheless, their *BRI1-BRLs* genes were still highly conserved in terms of protein sequences, gene numbers, and gene types, indicating that these results can be used as representative for most oaks.

It is important to determine why oak BRI1 selects the Asn-Gly-Ser-Ile (NGSI) sequence as the BR-binding region instead of the classical NGSM motif in most angiosperm BRI1s. In at least three oak BRI1s, the methionine in the NGSM motif was replaced by isoleucine. We also found that this amino acid mutation did not occur in *Fagus sylvatica* BRI1. Thus, we speculate that this amino acid mutation might have occurred after the divergence of *Quercus* and *Fagus*. Our transcriptional and genetic analyses showed that QmBRI1 can sense BR, which indirectly responds to the presence of its BR-binding ability. A previous study has shown that this amino acid mutation affected BR-binding ability [10]. However, the extent of the effect of this amino acid mutation in oak BRI1 on its BR-binding ability still needs to be further investigated by point mutation assays.

We found that oak BRI1s were closer to soybean BRI1s than to those of *Arabidopsis* and poplar. Consistent with previous reports, phylogenetic analysis revealed that the angiosperm *BRI1-BRLs* consisted of three major clades, with significant divergence between eudicots and monocots [11,33]. Additionally, in eudicot *BRI1-BRLs*, only the BRL2-type proteins did not diverge significantly between rosids and asterids, indicating that eudicot BRL2s may be more functionally conserved, e.g., that they are nonfunctional BR receptors [34]. Contrastingly, even BRI1-type proteins in rosids can still have high diversity, which is comparable to the results of a previous study [34]. The APG IV shows that rosids are primarily composed of fabids and malvids [26]. Oak and soybean come from the main clade below fabids, and their BRI1 proteins diverged significantly from those of other rosids (including poplar and *Arabidopsis*). Thus, the BRI1s of this clade may have evolved independently of most rosids.

We found that *QmBRI1* plays an important role as a *Q. mongolica* functional BR receptor in vascular patterning and xylem differentiation. In vascular tissues at different developmental stages, the expression patterns of two putative *Q. mongolica* functional BR receptors, *QmBRI1* and *QmBRL1*, were partially redundant; *QmBRI1* was highly expressed in young vascular tissues, comparable to the expression pattern of BR receptor genes in *Arabidopsis* and poplars [13,35,36]. We also noticed that the expression patterns of *BRI1-BRL1s* between the upper and basal stems of oak and poplar were completely opposite. The differences in the degree of lignification of the materials used in the two studies might explain this result [13]. Moreover, the *QmBRI1* and *QmBRL1* expression levels were inclined to increase first and then decrease over time during the early stage of BR-promoted xylem differentiation, consistent with the canonical BR negative feedback regulation mechanism [37]. To maintain BR homeostasis in plants, especially when the BR signaling output is strong, the canonical BR signal can be inhibited as a negative feedback loop, suppressing the expression of genes encoding positive signal components [37]. Therefore, both *QmBRI1* and *QmBRL1* are sensitive to exogenous BR and are regulated by double BR negative feedback, supporting their partially redundant receptor role as functional BR receptors in BR recognition or signaling [36]. Meanwhile, the relative expression of *QmBRI1* was higher than that of *QmBRL1* during this process, implying that *QmBRI1* plays an essential role in BR recognition and xylem differentiation. 

The ectopic expression of *QmBRI1* in *Arabidopsis* indicated that *QmBRI1* is a functional ortholog of *AtBRI1* and functions similarly to AtBRI1 in *Arabidopsis* growth and development. For example, overexpression of *QmBRI1* in the *bri1-5* mutant increased VB numbers and xylem differentiation, comparable to the finding that *Arabidopsis* transgenic lines (BRI1-GFP overexpression) showed more VBs and lignified tissues [31]. These results also supported the relatively high expression of *QmBRI1* in the *Q. mongolica* shoots, young stems, and BR-induced xylem differentiation. In addition, the function of the *QmBRL1* in secondary xylem development must be verified and the mechanism by which functional BR receptor genes in oak mediate BR signaling to regulate xylem differentiation must be further explored in the future.

## 4. Materials and Methods

### 4.1. Identification and Bioinformatics Analysis of BR Receptor Genes

The protein sequences of four *Arabidopsis BRI1-BRLs* genes (*AtBRI1*, *AtBRL1*, *AtBRL2*, and *AtBRL3*) [7] retrieved from The Arabidopsis Information Resource [38] were used as queries to obtain similar protein sequences in seven oaks (*Q. mongolica*, *Q. lobata*, *Q. suber*, *Q. robur*, *Q. gilva*, *Q. acutissima*, and *Q. variabilis*) and *F. sylvatica* genomes using the Protein Basic Local Alignment Search Tool (BLAST; E-value < 10^−20^). The obtained sequences were manually screened based on feature information obtained from the UniProtKB/Swiss-Prot database [39]. Screened protein sequences were submitted to the Expasy web portal [40] for further physicochemical property analysis. Only the BR receptor genes with complete protein sequences were retained. 

The identified oak BR receptor genes were subjected to bioinformatics analysis. Among them, the genome of *Q. suber* genome has not yet been assembled at the chromosomal level [19]. The chromosomal locations of *BRI1-BRLs* in *Q. mongolica*, *Q. lobata*, *Q. robur*, *Q. gilva*, and *Q. acutissima* were obtained from their genome annotation files [17,19,21,23,25]. The duplication events within *Q. mongolica* and *Q. lobata* genomes were detected by the Tbtools v1.108 internal program “One Step MCScanX” [41]. Gene structure analyses of *BRI1-BRLs* in three representative oak species (*Q. mongolica*, *Q. lobata*, and *Q. suber*) and *Arabidopsis* were performed using GSDS 2.0 [42]; and the subcellular localizations were predicted using Plant-mPLoc [43]. The signal peptides and transmembrane domains were predicted using SignalP 6.0 [44] and DeepTMHMM [45], respectively. Motif identification and analysis were performed using MEME-Suite 5.5.0 [46]. Using the same method as in the case of oaks, protein sequences that show similarity to *Arabidopsis BRI1-BRLs* were identified in *Selaginella moellendorffii* [47] as well as in eight representative angiosperm species (namely *Oryza sativa* [48], *Zea mays* [49], *Solanum lycopersicum* [50], *Glycine max* [51], *Populus trichocarpa* [52], *Aralia elata* [53], *Xanthoceras sorbifolium* [54], and *Tripterygium wilfordii* [55]). Among them, the protein sequence identified in *S. moellendorffii* was annotated as orthologs of EXCESS MICROSPOROCYTES 1 (EMS1). After aligning the BRI1-BRLs sequences of 12 species (*Arabidopsis*, the three oaks, and eight representative angiosperm species) and the sequences of EMS1 orthologs from *S. moellendorffii* using the Tbtools v1.108 internal program “MUSCLE Wrapper”, the results were trimmed using the “Quick Run TrimAL” program. Based on the lowest Bayesian information criterion, the best-fit model (JTT+I+G4) of the trimmed files was selected using ModelFinder v2.2.0 [56]. Maximum likelihood (ML) trees were then constructed using the Tbtools v1.108 internal program “IQ-TREE Wrapper” with 5000 bootstrap replicates. Bayesian inference trees were constructed using the PhyloSuite v1.2.3pre3 [57] internal program “Mrbayes”.

### 4.2. Vector Construction, Subcellular Localization, and Plant Transformation

The *AtBRI1* and *QmBRI1* coding sequences were obtained from the *Arabidopsis* and *Q. mongolica* cDNAs, respectively, using gene-specific primers (Appendix A). After being verified by sequencing, the stop codon was deleted from the *AtBRI1* and *QmBRI1* coding sequences, which were then fused in-frame to the N-terminus of green fluorescent protein (GFP) in the pRI101-GFP vectors containing the 35S promoter [58]. Two resulting plasmids and the control vector pRI101-GFP were individually transfected into *Arabidopsis* mesophyll protoplasts to evaluate the subcellular location of QmBRI1 [59]. Images of GFP fluorescence were captured using a C2-ER confocal laser scanning microscope (Nikon, Tokyo, Japan). As previously described [60], pRI101-QmBRI1-GFP was transfected with the *Arabidopsis bri1-5* mutant (Ws-2, Wassileskija-2 background) for complementation investigation. The transgenic lines were screened for 30 mg/L kanamycin resistance; homozygous lines were identified at the T3 generation using polymerase chain reaction (PCR) analysis.

### 4.3. Plant Growth Conditions, Histological Analysis, and Phenotypic Statistics

*Q. mongolica* seeds were collected from a genome-sequenced tree at Shenyang Agricultural University, Shenyang, Liaoning, China. Surface-sterilized seeds sprouted in non-woven bags (height: 25 cm; diameter: 13 cm) that included peat, perlite, and vermiculite (3:1:1, *v*/*v*/*v*) under the following requirements: 16/8 h light/dark cycle and 27/25 °C light/dark temperature. *Q. mongolica* seedlings were watered once a week with 50% Hoagland’s solution [61]. 

*Arabidopsis* seeds including the mutant *bri1-5* (Ws-2, Wassileskija-2 background) [27], ecotype Ws-2, and *QmBRI1* transgenic lines were surface-sterilized with ethanol and sodium hypochlorite solution, then washed with sterilized distilled water and planted on 1/2 Murashige–Skoog (MS) media. After low-temperature stratification, the seeds were cultivated in an incubator under the following conditions: 16/8 h light/dark cycle and 23/22 °C light/dark temperature. After the seedlings had developed true leaves, they were planted in soil for culturing [62]. After 21 days of germination, the *Arabidopsis* leaf characteristics and petiole lengths were observed and measured. Furthermore, after 35 days of germination, the *Arabidopsis* inflorescence stem heights and silique lengths were measured.

A mixture of 24-epibrassinolide (BL; Solarbio, Beijing, China) and lanolin (Solarbio, Beijing, China) was prepared according to the following these steps: BL powder was dissolved in alcohol and prepared as the BL mother solution; lanolin was then placed in a container and melted in a water bath at 55 °C; the BL mother solution was added to the container to prepare a mixture of BL and lanolin at 1 μM concentration; the mixture was stirred until the ethanol completely evaporated; finally, it was cooled at 18 °C for subsequent use. Using pure lanolin as a control, 250 μL of the mixture was applied to the epidermis of local basal stems of 2-month-old *Q. mongolica* seedlings (3 cm long). Twenty-eight days after the treatment, *Q. mongolica* local stems were harvested and fixed in classical FAA fixative for 24 h, and then embedded in paraffin until further use. The main inflorescence stem bases of the Ws-2 ecotype, *bri1-5* mutant, and transgenic line were collected 35 days after germination, and fixed and embedded using the same method. The paraffin-embedded samples of *Q. mongolica* and *Arabidopsis* samples were cut into 7–10 μm-thick pieces using an RM2255 rotary microtome (Leica, Wetzlar, Germany). After paraffin removal [63,64], *Q. mongolica* sections were double-stained with Safranin O and Fast Green, whereas *Arabidopsis* sections were stained with 0.05% toluidine blue. All mounted sections were photographed under an ECLIPSE Ci-L upright light microscope (Nikon, Tokyo, Japan).

ImageJ 1.53k software [65] was used to measure the width of the newly developed *Q. mongolica* secondary xylem and *Arabidopsis* leaf and silique lengths, inflorescence stem height, fascicular xylem radius, and interfascicular arc middle part and bundle-flanking region lengths.

### 4.4. RNA Extraction and qRT-PCR Analysis

The roots, young leaves, shoots, the phloem and xylem of the upper stems (first stem internode), and the phloem and xylem of the basal stems (second stem internode) were collected from 2-month-old *Q. mongolica* seedlings. The local basal stems were collected at 0, 6, 12, 24, 36, and 48 h after the BL and blank treatments. Leaves of *Arabidopsis* lines (*bri1-5* mutants, Ws-2 ecotypes, *QmBRI1OX-1* lines, and *QmBRI1OX-3* lines) were collected 21 days after germination. All samples were immediately frozen in liquid nitrogen after collection. Total RNA from *Q. mongolica* and *Arabidopsis* was extracted using an RNAprep Pure Plant Kit (Tiangen, Beijing, China). The PrimeScript™ II first-strand cDNA Synthesis Kit (Takara, Dalian, China) was used to obtain the cDNA. SuperReal PreMix Plus (Tiangen, Beijing, China) was used to perform quantitative real-time PCR (qRT-PCR) using an ABI StepOnePlus machine (Applied Biosystems, Foster City, CA, USA). *Ubiquitin* and *Actin2* were utilized as internal reference genes for *Q. mongolica* and *Arabidopsis*, respectively; the gene-specific primers for qRT-PCR are listed in Appendix A. The gene expression levels were determined by using the 2^−△△CT^ algorithm [66].

## 5. Conclusions

We performed a genome-wide identification of three types of classical BR receptor genes in three representative oak species. Next, we analyzed their chromosome distribution, replication, structure, protein properties, conserved domains and motifs, BR-binding regions, and evolution. The *BRI1s* and *BRL1s* in them were finally identified as oak functional BR receptor genes. We found that the classical BR-binding region (NGSM motif) in plant functional BR receptors was not conserved in oak BRI1s. The methionine in NGSM motif in oak BRI1s was replaced by isoleucine, and this amino acid mutation occurred after the divergence of *Quercus* and *Fagus*. BRI1s, a subclade species of fabids to which oaks belong, evolved differently from most rosids. Additionally, we analyzed the spatiotemporal expression patterns of functional BR receptors in *Q. mongolica* and performed functional verification on *QmBRI1*. QmBRI1 was localized to the plasma membrane, and it played an important role in young vascular tissue development, especially in vascular patterning and xylem differentiation. These findings serve as an important complement to the findings of the structural, functional and evolutionary studies on functional BR receptors in woody plants and provide an example of natural mutation occurring in the classical BR-binding region of angiosperm BRI1s.

## Figures and Tables

**Figure 1 ijms-24-16405-f001:**
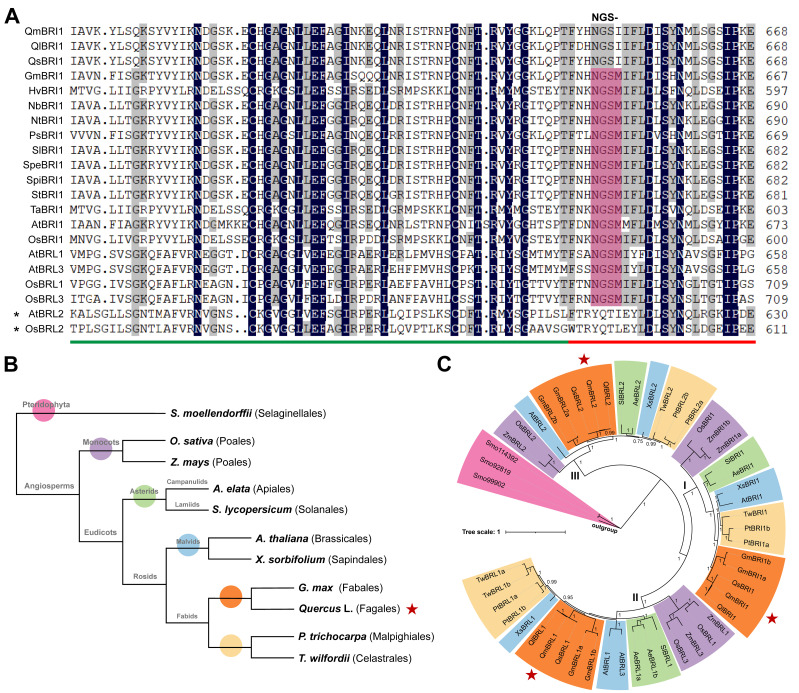
Multiple sequence alignment of ID-LRR domains and phylogenetic analysis of *BRI1-BRLs*. (**A**) Only the protein sequences of *BRI1-BRLs* genes with strong literature support in the UniProt Knowledgebase were selected for their ID-LRR domain alignment. These protein sequences are (numbers in parenthesis indicate UniProtKB entries): GmBRI1 (*Glycine max*; C6FF79); HvBRI1 (*Hordeum vulgare*; Q76CZ4); NbBRI1 (*Nicotiana benthamiana*; A4LAP7); NtBRI1 (*Nicotiana tabacum*; A6N8J1); PsBRI1 (*Pisum sativum*; Q76FZ8); SlBRI1 (*Solanum lycopersicum*; F2XYF6); SpeBRI1 (*Solanum peruvianum*; Q8L899); SpiBRI1 (*Solanum pimpinellifolium*; A4LAP5); StBRI1 (*Solanum tuberosum*; A4LAP6); TaBRI1 (*Triticum aestivum*; Q0ZA03); AtBRI1 (*Arabidopsis thaliana*; O22476); OsBRI1 (*Oryza sativa*; Q942F3); AtBRL1 (Q9ZWC8); AtBRL3 (Q9LJF3); OsBRL1 (Q69JN6); OsBRL3 (Q6ZCZ2); AtBRL2 (Q9ZPS9); OsBRL2 (Q7G768). Green line indicates the ID; red line, an LRR. Amino acids highlighted in dark blue are identical in all sequences; in gray are identical in most sequences; in pink are the canonical NGSM motif regions. Asterisked sequences are controls for other sequences. (**B**) Schematic diagram of the taxonomic relationship between oaks, *Selaginella moellendorffii*, and nine representative angiosperms (*Oryza sativa*, *Zea mays*, *Aralia elata*, *Solanum lycopersicum*, *Xanthoceras sorbifolium*, *Glycine max*, *Populus trichocarpa*, *Arabidopsis thaliana*, and *Tripterygium wilfordii*). (**C**) Phylogenetic tree was constructed using the Bayesian method. The homologs of EMS1 from *Selaginella moellendorffii* were used as an outgroup. Only probabilities greater than 0.7 are displayed as numbers above the branches, which represent posterior probability values. The star symbol represents the oaks group. The different color blocks represent the different plant groups.

**Figure 2 ijms-24-16405-f002:**
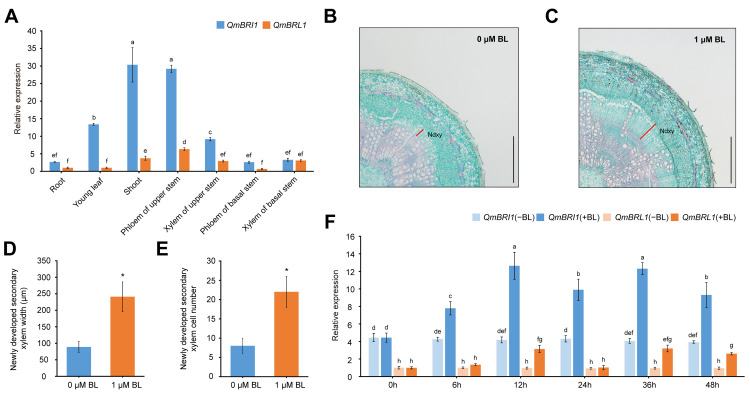
Expression patterns of functional BR receptor genes in *Q. mongolica*. (**A**) *QmBRI1* and *QmBRL1* expression patterns in various tissues of 2-month-old *Q. mongolica* seedlings. The different lowercase letters indicate significant differences (*p* < 0.05, one-way ANOVA with Duncan’s significant difference test). (**B**,**C**) Wood histology of local stems in *Q. mongolica* seedlings. Epidermis of local stems of 2-month-old *Q. mongolica* were treated with 0 and 1 μM 24-epibrassinolide (BL) treatments for 28 days. Ndxy, newly developed secondary xylem. Scale bar = 500 μm. (**D**) Newly developed secondary xylem width. (**E**) Newly developed secondary xylem cell number. Asterisks indicate significant differences compared with 0 μM BL-treated group (* *p* < 0.05, Student’s *t*-test, *n* = 3). (**F**) *QmBRI1* and *QmBRL1* expression patterns in BR-induced xylem differentiation. −BL, 0 μM BL-treated group; +BL, 1 μM BL-treated group. The *Q. mongolica UBQ10* gene was used as an internal control. The different lowercase letters indicate significant differences (*p* < 0.05, one-way ANOVA with Duncan’s significant difference test). Error bars represent ± SD from three biological repeats.

**Figure 3 ijms-24-16405-f003:**
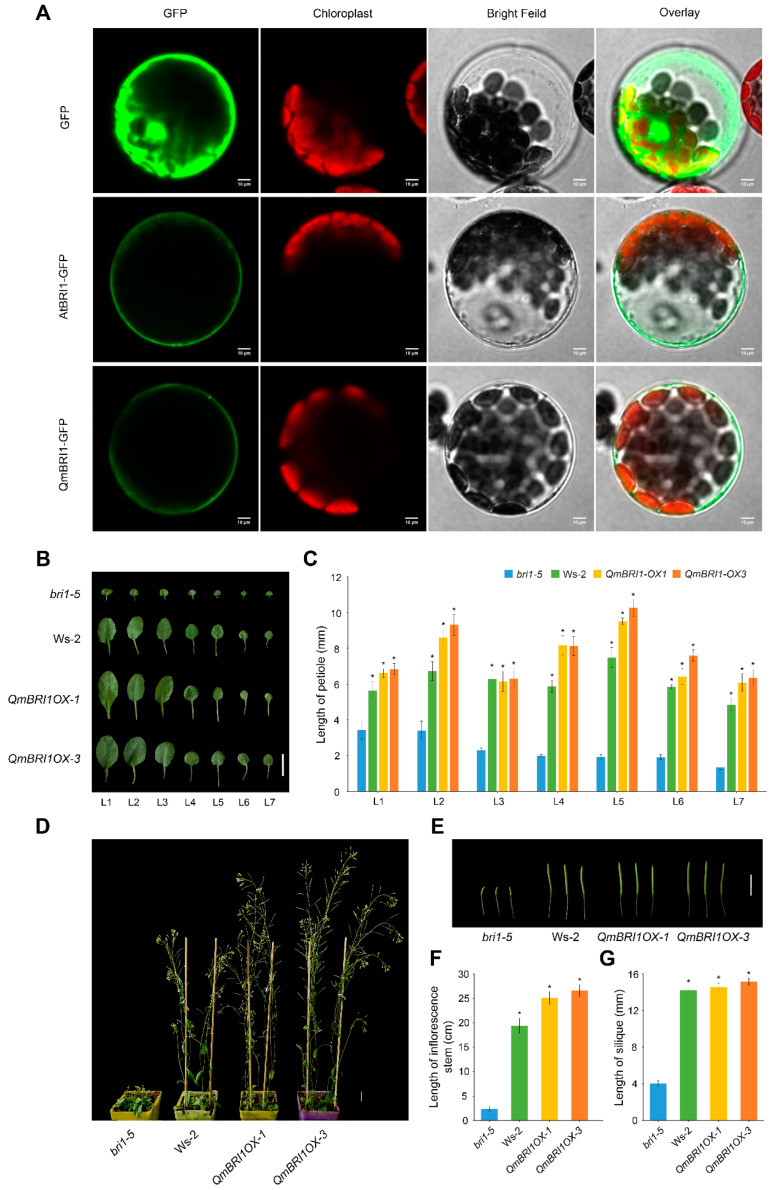
Protein sub-cellular localization and mutant complementation analyses. (**A**) QmBRI1 is localized to the cell membrane. The control pRI101-GFP, pRI101-AtBRI1-GFP, and pRI101-QmBRI1-GFP vectors were transiently expressed in *Arabidopsis* mesophyll cell protoplasts. The fluorescence was examined using a confocal laser scanning microscope. Scale bars = 10 μm. (**B**) *QmBRI1* overexpression rescued the growth defects of leaves and petioles in *Arabidopsis bri1-5* mutants. Leaf phenotypes of 3-week-old *Arabidopsis bri1-5* mutant, Ws-2 ecotype, and two transgenic *bri1-5* plants complemented with *QmBRI1*. Scale bar = 1 cm. (**C**) Petiole lengths of 3-week-old *Arabidopsis bri1-5* mutant, Ws-2 ecotype, and two transgenic *bri1-5* plants complemented with *QmBRI1*. Error bars represent ± SD from five biological repeats. Asterisks indicate significant differences compared with *bri1-5* mutants (* *p* < 0.05, Student’s *t*-test, *n* = 5). (**D**,**E**) *QmBRI1* overexpression rescued the growth defects of inflorescence stems and siliques in *Arabidopsis bri1-5* mutants. Plant and silique phenotypes of 5-week-old *Arabidopsis bri1-5* mutant, Ws-2 ecotype, and two transgenic *bri1-5* plants complemented with *QmBRI1*. Scale bar = 1 cm. (**F**,**G**) Heights and silique lengths of 5-week-old *Arabidopsis bri1-5* mutant, Ws-2 ecotype, and two transgenic *bri1-5* plants complemented with *QmBRI1*. Error bars represent ± SD from 15 biological repeats. Asterisks indicate significant differences compared with *bri1-5* mutants (* *p* < 0.05, Student’s *t*-test, *n* = 15).

**Figure 4 ijms-24-16405-f004:**
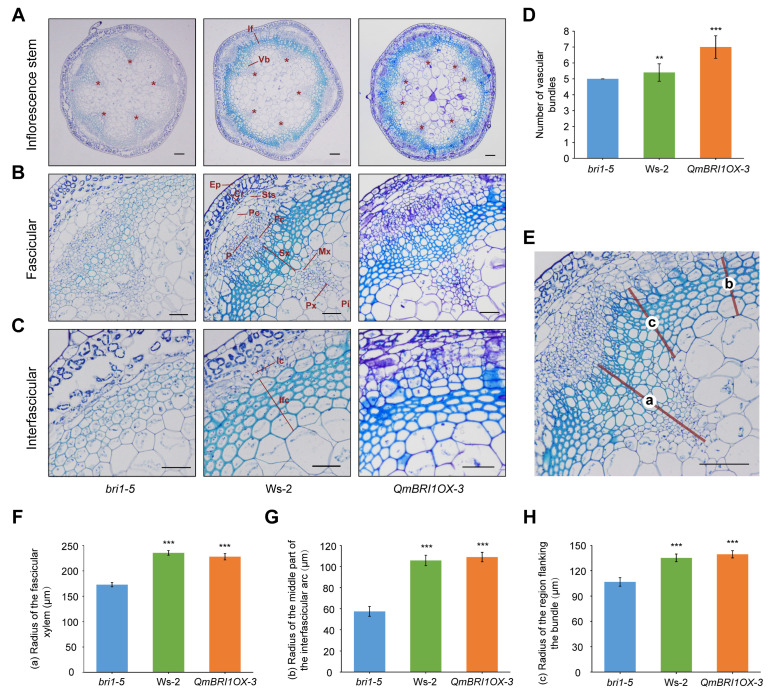
Overexpression of *QmBRI1* in *bri1-5* mutants increased vascular bundle (VB) numbers and xylem differentiation. (**A**) The representative cross section images of the inflorescence stems of 5-week-old *Arabidopsis bri1-5* mutant, Ws-2 ecotype, and transgenic *bri1-5* plant complemented with *QmBRI1*. *, vascular bundle. VB, vascular bundle; IF, interfascicular. Scale bar = 100 μm. (**B**) The representative cross section image of inflorescence stem fascicular of 5-week-old indicated genotypes. Fc, fascicular cambium; Sx, secondary xylem; Mx, metaxylem; Px, protoxylem. Scale bar = 100 μm. (**C**) The representative cross section image of inflorescence stem interfascicular of 5-week-old indicated genotypes. Ic, interfascicular cambium; Ifc, interfascicular fiber cell. Scale bar = 50 μm. (**D**) Statistics of VB number in the inflorescence stem of 5-week-old indicated genotypes. Error bars represent ± SD from five biological repeats. Asterisks indicate significant differences compared with *bri1-5* mutants (** *p* < 0.01, *** *p* < 0.001, Student’s *t*-test, *n* = 5). (**E**) A schematic diagram of the production of lignified tissue in Ws-2 ecotype. a, fascicular xylem; b, the middle part of interfascicular arc; c, bundle-flanking region. Scale bar = 200 μm. (**F**–**H**) Radii of the fascicular xylem, the interfascicular arc middle part, and the bundle-flanking region in 5-week-old indicated genotypes. Error bars represent ± SD from five biological repeats. Asterisks indicate significant differences compared with *bri1-5* mutants (*** *p* < 0.001, Student’s *t*-test, *n* = 3).

**Table 1 ijms-24-16405-t001:** General information about the BR receptor genes in three representative oak species based on bioinformatics analysis.

Name	Gene ID	AA (aa)	SP (aa)	TM (aa)	Mw (kDa)	pI	II	AI	GRAVY
*Quercus mongolica*									
*QmBRI1*	Qm025300	1189	1–24	791–809	130.09	6.07	37.50	98.40	−0.048
*QmBRL1*	Qm013727	1221	1–37	835–855	132.37	5.77	35.94	97.67	−0.002
*QmBRL2*	Qm012212	1133	1–29	755–775	124.57	6.01	31.92	101.21	−0.062
*Quercus lobata*									
*QlBRI1*	QL06p031490	1189	1–24	791–809	129.94	5.99	36.76	98.32	−0.046
*QlBRL1*	QL08p000343	1221	1–37	835–855	132.34	5.82	36.46	98.07	0.001
*QlBRL2*	QL03p039295	1133	1–29	755–775	124.50	5.94	32.33	101.21	−0.058
*Quercus suber*									
*QsBRI1*	XP_023879012.1	1189	1–24	791–809	129.82	5.91	36.44	97.34	−0.051
*QsBRL1*	XP_023887804.1	1221	1–37	835–855	132.42	5.66	36.66	98.30	−0.000
*QsBRL2*	XP_023912129.1	1133	1–29	755–775	124.46	5.94	31.77	100.94	−0.067

AA, amino acid number; SP, the position of signal peptide; TM, the position of transmembrane helix; Mw, molecular weight; pI, theoretical isoelectric point; II, instability index; AI, aliphatic index; GRAVY, grand average of hydrophobicity.

## Data Availability

Publicly available datasets were analyzed in this study. Accession numbers can be found in the Section 4.

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
