# Peer review of "Identification of Functional Brassinosteroid Receptor Genes in Oaks and Functional Analysis of QmBRI1"

_ijms, 2023, doi:10.3390/ijms242216405_

Round 1
Reviewer 1 Report
Comments and Suggestions for Authors
The manuscript appears to have been written competently, but some modifications could be made to improve the clarity and readability of readers. The comments have been provided to
suggest areas where revisions could be made to enhance the quality of the manuscript and make it more accessible to broader readers and researchers (Please find the attached manuscript file with highlighted text and comments).

Reviewer 2 Report
Comments and Suggestions for Authors
The ms IJMS-2684956 entitled “Identification of Functional Brassinosteroid Receptor Genes in Oaks and Functional Analysis of QmBRI1” studies nine BR receptors in three representative oak species. In oak BRI1s, the methionine in the 15 conserved Asn-Gly-Ser-Met (NGSM) motif was replaced by isoleucine. 16 QmBRI1 was relatively highly expressed during BR-induced xylem differentiation and in young leaves, shoots, and the phloem and xylem of young stems of Quercus mongolica while QmBRL1 not. The important role of QmBRI1 in oak growth and development, especially in vascular patterning and xylem differentiation was assessed in Arabidopsis plants. QmBRI1 and AtBRI1 differed in their stem vascular development regulation strategies by mediating the BR signaling-induced expression of HD-ZIP III transcription factors.
The ms is well written and the work is well executed nevertheless few points should be clarified. The development of oaks is different from that of Arabidopsis, and this fact could be also reflected by the induction of BRI1 genes in the presence of BL. Another major point is the use of overexpressors (most likely by 35S). To assess the functional complementation of an orthologous protein most of the times either the promoter of the endogenous gene (the one that will be complemented) should be used or the original (the one that is carried by the gene that will be transferred). The overexpressors sometimes does create mutations or it does not give the tissue specificity for a correct expression. And this is obvious. How come a 35S promoter is regulated by BL (see Fig. 4)? Some of the results from Fig. 5 are coming because of o/e or because of the activity of the QmBRI1 itself?
Round 2
Reviewer 2 Report
Comments and Suggestions for Authors
Changes have improved the ms. However, the authors should avoid any conclusion or comments about the 35S promoter being regulated by BL. The authors should also avoid any link between the o/e lines the application of BL (in molecular and physiological studies). Otherwise it means that the 35S (and the gene that is fused with the promoter) is regulated by BL. It may be regulated, but we do not know, as yet. Besides the 35S may be regulated not directly but indirectly.
And a last one, if some other authors did use the 35S for their studies, it does not necessarily mean that it is correct and by no means does it provide a "passport” or excuses to other scientists to follow the same path. It would be wiser for all scientists to take the time and think thoroughly about the work that they will conduct.
Author Response
请参阅附件
